# Robust H-K Curvature Map Matching for Patient-to-CT Registration in Neurosurgical Navigation Systems

**DOI:** 10.3390/s23104903

**Published:** 2023-05-19

**Authors:** Ki Hoon Kwon, Min Young Kim

**Affiliations:** 1School of Electronic and Electrical Engineering, Kyungpook National University, Daegu 41566, Republic of Korea; kwons149@naver.com; 2Research Center for Neurosurgical Robotic System, Kyungpook National University, Daegu 41566, Republic of Korea

**Keywords:** H-K curvature, image-to-patient registration, spherical unwrapping, iterative closest point (ICP), template matching

## Abstract

Image-to-patient registration is a coordinate system matching process between real patients and medical images to actively utilize medical images such as computed tomography (CT) during surgery. This paper mainly deals with a markerless method utilizing scan data of patients and 3D data from CT images. The 3D surface data of the patient are registered to CT data using computer-based optimization methods such as iterative closest point (ICP) algorithms. However, if a proper initial location is not set up, the conventional ICP algorithm has the disadvantages that it takes a long converging time and also suffers from the local minimum problem during the process. We propose an automatic and robust 3D data registration method that can accurately find a proper initial location for the ICP algorithm using curvature matching. The proposed method finds and extracts the matching area for 3D registration by converting 3D CT data and 3D scan data to 2D curvature images and by performing curvature matching between them. Curvature features have characteristics that are robust to translation, rotation, and even some deformation. The proposed image-to-patient registration is implemented with the precise 3D registration of the extracted partial 3D CT data and the patient’s scan data using the ICP algorithm.

## 1. Introduction

Image-guided surgery is a technology that helps doctors perform accurate surgical procedures by using augmented reality techniques to match the medical image of the surgical site to the actual surgical site [1,2,3,4]. Unlike conventional surgical methods that rely only on the experience and knowledge of the surgeons, it is possible to assist the operation by utilizing the image information of the patient’s surgical area in real time. In this system, the surgeon can operate on the patient while looking at the medical image matched with the surgical site on the monitor as well as the actual surgical site. A surgical navigation system is one method of image-guided surgery [5,6,7,8]. A surgical navigation system provides optimum trajectory to the surgical target, just as a vehicle navigation system facilitates the driver with information on the map and route to the destination. This system informs the location of the surgical site, the current location of the surgical tool, and whether the surgical tool is safely approaching the target lesion on a medical image. To implement this system, the relative positional relationship between the surgical site and the surgical tool must be tracked in real time. The coordinates of the surgical tool can be displayed in real time on the matched medical image using the obtained position and attitude information of the surgical tool. Therefore, image-to-patient registration [9,10,11], which is a coordinate system matching process between medical images, such as computerized tomography (CT) or magnetic resonance imaging (MRI), and real patient coordinates before surgery, is essential for an accurate surgical navigation system.

In the image-to-patient registration process, two different coordinates of the medical image and the actual patient are transformed into one coordinate system. Paired-point registration is one of the typical image-to-patient registration methods [12]. It utilizes corresponding points in the patient and medical image to match the patient’s CT/MRI coordinate system with the patient’s world coordinate system. If at least three points that correspond to each other between two 3D data are known, the rotation and translation information between the two data can be obtained for matching by calculating the relation between these corresponding points [13]. Skin-attached fiducial markers are usually used to obtain these corresponding points [14,15]. In a pre-operative process, fiducial markers are attached to the patient; then, a medical image is captured. Next, image-to-patient registration is performed in the operating room between the patient with the attached fiducial markers and the medical image. However, marker-based methods are inconvenient to use. In addition, if the pose of the marker attached to the patient when obtaining the CT or MRI image and the pose during the operation differs, matching error can also increase. To solve these problems, 3D data-based methods have been proposed [16]. They utilize the 3D surface measurement data of a patient to perform image-to-patient registration without fiducial markers [17]. In this method, a 3D surface measurement sensor is used to obtain the 3D surface data of the patient [18,19]. Then, image-to-patient registration is implemented by matching the 3D surface data with the corresponding part of the 3D data converted from the medical image. It is possible to project the CT/MRI data onto the world coordinate system during surgery by matching the 3D surface data of the patient with the CT/MRI data. After image-to-patient registration is completed, when the patient moves, the optical tracker keeps track of the marker for tracking attached to the patient rather than performing patient registration again. Therefore, the registration state can be continuously maintained by using the posture information of the tracked marker.

A 3D data precision matching algorithm is essential to transforming a patient’s 3D surface data and 3D CT/MRI data into a final coordinate system. Among several 3D matching algorithms, the iterative closest point (ICP) algorithm is the most representative and widely used one [20,21]. The ICP algorithm repeats the process of defining the closest points between two 3D data as corresponding points and minimizing the sum of the distances between the corresponding points for precision data registration. Conventional ICP algorithms without a proper initial location involve a lot of computation and can also suffer from incorrect matching results due to the local minimum problem [22]. The 3D data registration process can become faster and more accurate if a proper initial location is provided before performing the ICP algorithm. In this paper, a 3D registration method based on curvature matching to automatically find the proper initial location for the ICP algorithm is proposed in the head region for neurosurgery. This paper is an expanded version of the conference paper [23], and the basic concept can also be found in the conference paper. The proposed method utilizes the natural features of the skin surface, such as the nose and ears, eliminating the need for fiducial markers. The natural feature data are common between the patient and the CT/MRI image, and due to the relatively large rate of change in the data, these are easy to distinguish from other parts. Therefore, registration and matching errors can be reduced using these feature data. The proposed method finds the proper initial location for the ICP by converting 3D data to 2D curvature images and automatically performing curvature image matching. The proposed matching process is based on the characteristic that curvature features are robust to rotation, translation, and even some deformation [24,25]. To implement image-to-patient registration, the ICP algorithm between 3D CT data and 3D scan data is employed by utilizing curvature matching region information rather than the complicated 3D template matching methods [26,27].

## 2. Proposed Patient-to-CT Registration Method

In this paper, an automatic and robust image-to-patient registration method is proposed for neurosurgery using curvature map conversion and matching. Three-dimensional surface data to match CT/MRI data are obtained using a surface measurement sensor by measuring natural feature surfaces such as the nose, eyes, and ears on a patient’s head. The proposed registration method can avoid the local minimum problem, because suitable initial positions in the CT surface data to match the 3D surface data are automatically found. Figure 1 displays the schematic diagram of the proposed algorithm, which consists of three steps. First, the CT surface data of the patient’s head are transformed into a 2D image using spherical unwrapping, since the head’s surface data are similar to a sphere [28]. Similarly, the surface measurement data obtained with the sensor are also converted to a 2D image using spherical unwrapping and the mean radius computed from the CT surface data. In the second step, both 2D images are converted to H-K curvature images by calculating the partial differentiation of the peripheral pixel intensity.

After converting the H-K curvature image, the image matching process between two curvature images is performed. In the final step, a 3D region of interest (ROI) from the CT surface data is extracted by utilizing the curvature matching points obtained in the previous step. Since the curvature image is a projection of 3D data onto a plane, the 2D points can be converted into 3D coordinates with an inverse operation. The proper initial coordinates for running the ICP algorithm can be estimated on the CT surface data by inversely mapping the corresponding 2D coordinates to 3D coordinates. Automatic image-to-patient registration is implemented by matching the CT ROI from CT surface data and 3D surface data using the ICP algorithm.

### 2.1. Mapping 3D CT Data to 2D Image

A CT image is a set of 2D cross-section images obtained by repeatedly scanning a patient along the Z-axis. By aligning the corresponding Z-axis of the cross-section images and collecting all the points, 3D point cloud data can be reconstructed from the CT image, as shown in Figure 1, which shows the 3D CT data of a head phantom. The points for mapping to the 2D Image in the 3D CT data should be skin surface data, which are the outermost points. Therefore, a method is required to extract these points from the entire 3D CT data. To perform this, the 3D CT data are divided into the X-axis and Y-axis, and the outermost points are extracted based on each axis. By dividing the 3D CT data in half along the Y-axis at the center of 3D CT data, the outermost points of 3D CT data on the Y-axis are extracted with the maximum value operation on the left side of the separated 3D CT data and the minimum value operation on the right side of the separated 3D CT data. To increase the precision of the data, similarly, after dividing the 3D CT data in half along the X-axis at the center of 3D CT data, the outermost points of 3D CT data on the X-axis are also extracted with the maximum value operation on the front side of the separated 3D CT data and the minimum value operation on the back side of the separated 3D CT data. The outermost points of the total 3D CT data, which are skin surface data, are obtained by merging these two sets of extracted data.

The 3D shape of the CT data extracted from the patient’s head surface is roughly spherical. Therefore, it is more effective to convert the entire 3D CT data of the skin surface into a 2D image using spherical coordinate system conversion, called spherical unwrapping, rather than a simple plane transformation. According to a principle similar to equirectangular projection, which transforms the map of a globe into a 2D plane map, this method converts every point of spherical 3D surface data into a 2D depth image. Spherical unwrapping can be performed along any axis of the 3D data, and when the patient faces in the direction of the Y-axis in 3D CT data, spherical unwrapping based on the Y-axis is given by the following equations: (1)$$r_{i}=\sqrt{x_{i}^{2}+y_{i}^{2}+z_{i}^{2}}$$
(2)$$\varphi_{i}=\arctan\left(\frac{y_{i}}{x_{i}}\right)$$
(3)$$\theta_{i}=\arcsin\left(\frac{z_{i}}{r_{i}}\right)$$

The center coordinates of the 3D CT data are obtained by calculating the average coordinates of all the 3D CT data points. Assuming that this center coordinates represent the center of a sphere, the 3D CT data are translated in parallel so that the origin of the coordinate system is located at the center of the 3D CT data to perform spherical coordinate system transformations. The process of mapping 3D CT data to a 2D image is performed using Equations (Equation 1)–(Equation 3). As shown in Equation (Equation 1), distance $r_{i}$ between the center point (origin) and each point $p_{i}$ of 3D CT data is used as the image intensity value. $\varphi_{i}$, obtained from the X- and Y-coordinate values of each point $p_{i}$, is the width coordinate of the image, and $\theta_{i}$, obtained from the Z-coordinate value and $r_{i}$ value of each point $p_{i}$, is the height coordinate of the image, as shown in Equations (Equation 2) and (Equation 3). All $r_{i}$ values are mapped to the calculated width and height coordinates of the transformed image. This image projection process shows how points in the 3D CT data are transformed into a 2D image in Figure 2 using the spherical unwrapping conversion equation. To obtain an unwrapped CT image at the desired resolution, φ and θ are multiplied by the target resolution value, respectively. Figure 2 also shows the result of spherical unwrapping on 3D CT data, and the unwrapped CT image is a one-channel depth image.

### 2.2. Mapping 3D Scan Data to 2D Image

After converting the 3D CT data to a 2D image, the next step is to convert the 3D scan data obtained with a surface measurement sensor to a 2D image for image matching. The 3D scan data are gathered by projecting a structured light pattern on the surface of the target and analyzing the output of the patterns. In order to achieve accurate results of image matching, the two images to be matched should be on a similar scale. Therefore, the image acquired by mapping the 3D scan data should have a scale similar to that of the unwrapped CT image. Since the 3D scan data are only a partial measurement of the facial surface, they are flat data, unlike 3D CT data, which are spherical. Therefore, it is possible to map the 3D scan data to a 2D image using a simple planar projection, but it is not easy to set a scale similar to that of the unwrapped CT image previously obtained.

To address this issue, the 3D scan data are also mapped to the 2D image using spherical unwrapping and other techniques. Although the 3D scan data are planar, they can be aligned with the axis of the 3D CT data, and the $r_{mean}$ of the 3D CT data, as the average radius of the sphere, can be used to roughly place them on the spherical surface of the 3D CT data, as shown in Figure 3. In the same way, 3D scan data can also be mapped to the 2D image using spherical coordinate system conversion. Once the 3D scan data are placed on the spherical surface of the 3D CT data, spherical unwrapping is performed to generate an image on a scale similar to that of the unwrapped CT image. The 3D scan data are the partial area data of the head, and they are only mapped to a portion of the whole image. To ensure accurate image matching, only the area where the 3D scan data are mapped is extracted as a template. Figure 3 shows the results of template extraction in the unwrapped 3D scan image corresponding to the nose and right ear.

### 2.3. H-K Curvature Image Conversion

Curvature refers to the rate of change indicating the extent to which a curve or a curved surface deviates from a flat plane. Principal curvature refers to the maximum and minimum curvatures among the curvatures on a curved surface. Mean (H) curvature represents the average value of principal curvature, while Gaussian (K) curvature represents the product value of principal curvature. Both curvatures are commonly used for surface shape classification [29]. Since these curvatures are only determined by the surface form, such curvature features are robust to rotation, translation, and even some deformation. As the depth values of 3D scan data can be primarily affected by noise and variations in head poses during scanning, a robust image using a characteristic of curvature is necessary. The intensity of the unwrapped image for 3D CT data and 3D scan data represents the distance between the origin of the coordinate system and the corresponding point. Thus, this value can represent the surface shape of 3D data, and the H curvature and K curvature of the corresponding image coordinates can be obtained by calculating the partial differentiation using the N × M mask operation on the image intensity [30]. This method allows robust curvature features to be extracted from the unwrapped image.
(4)$$g_{ij}\left(x,y\right)=a_{ij}+b_{ij}\left(x-x_{i}\right)+c_{ij}\left(y-y_{j}\right)+d_{ij}\left(x-x_{i}\right)\left(y-y_{j}\right)+e_{ij}{\left(x-x_{i}\right)}^{2}+f_{ij}{\left(y-y_{j}\right)}^{2},\;\left(i=1-N,j=1-M\right)$$
(5)$$f_{x}\left(x_{i},y_{j}\right)=b_{ij},\;f_{y}\left(x_{i},y_{j}\right)=c_{ij},\;f_{xy}\left(x_{i},y_{j}\right)=d_{ij},\;f_{xx}\left(x_{i},y_{j}\right)=2e_{ij},\;f_{yy}\left(x_{i},y_{j}\right)=2f_{ij}$$
(6)$$H\left(x,y\right)=\frac{\left(1+f_{y}^{2}\right)f_{xx}-2f_{x}f_{y}f_{xy}+\left(1+f_{x}^{2}\right)f_{yy}}{2{\left(1+f_{x}^{2}+f_{y}^{2}\right)}^{3/2}}$$
(7)$$K\left(x,y\right)=\frac{f_{xx}f_{yy}-f_{xy}^{2}}{2{\left(1+f_{x}^{2}+f_{y}^{2}\right)}^{2}}$$

Equations (Equation 4)–(Equation 7) describe the process of calculating H-K curvatures in the depth image. Equation (Equation 4) is a biquadratic polynomial equation that uses the intensities of the surrounding image. N and M represent the mask size, while x and y denote the coordinates of the image for which the curvature is to be obtained. To calculate the curvature, the biquadratic polynomial equation, Equation (Equation 4), is obtained by performing an N × M mask operation around the image coordinate, and the coefficients of Equation (Equation 4) are calculated using least squares fitting. By substituting the coefficients of Equation (Equation 4) into Equation (Equation 5), the H and K curvatures of the corresponding coordinates can be obtained using Equations (Equation 6) and (Equation 7), respectively. These curvature values change dramatically in natural features such as eyes, nose, and ears while remaining constant in other areas with less variation, which can further emphasize the features. To utilize the characteristics of natural features for image matching, both images from unwrapped 3D CT data and 3D data are converted to H-K curvature images with H-K curvature values as image intensity.

The H-K curvature images of the 3D CT data and 3D scan data are visualized as 3D meshes in Figure 4 and Figure 5, respectively. The reason for showing the curvature image as a mesh is that the curvature value is too small, and mesh representation shows the relative variance of the surrounding value better than the regular image. As shown in both figures, natural features such as nose and ears are emphasized in H-K curvature images, as these areas have distinct curvature values compared with the surrounding areas.

### 2.4. Curvature Image Matching

To perform image matching between two curvature images acquired from CT data and scan data, normalized cross-correlation (NCC) is used [31]. In signal processing, cross-correlation is a method for measuring the similarity of two waveforms by shifting one waveform relatively to the other. To apply cross-correlation in image processing for measuring the similarity between two images, first, the images should be normalized by subtracting the mean and dividing by the standard deviation. This method ensures that the correlation measure is independent of differences in the absolute values of the image intensities. Equation (Equation 8) shows the formula for calculating NCC for image matching.
(8)$$\gamma \left(u,v\right)=\frac{\Sigma_{x,y}\left[f\left(x,y\right)-{\bar{f}}_{u,v}\right]\left[t\left(x-u,y-v\right)-\bar{t}\right]}{{\left\{\Sigma_{x,y}{\left[f\left(x,y\right)-{\bar{f}}_{u,v}\right]}^{2}\Sigma_{x,y}{\left[t\left(x-u,y-v\right)-\bar{t}\right]}^{2}\right\}}^{0.5}}$$

The process of image matching involves measuring linear variations and geometric similarity between two images by shifting the smallest image as a template, creating a relationship map between the two images, and selecting the largest values as the matching points. NCC is a method used to estimate the correlation between two images using normalization. It is independent of linear differences between the intensities of both images and is less influenced by the absolute values of image intensity. Thus, NCC is appropriate for matching relative shapes and is utilized for curvature image matching in this research. Since NCC is a kind of template matching, one image must be chosen as the template. In this case, the curvature image of the 3D scan data, which is smaller than the curvature image of the 3D CT data, is selected as the template image for curvature image matching. As shown in Figure 6, NCC is utilized to match the curvature images of the 3D scan data and CT data. Matching values for each datum are separately obtained using the H and K curvature images, and the coordinates with the largest average of the matching values of each datum are regarded as matching coordinates. The results of curvature image matching are displayed in Figure 6, where the matching area is shown by a bounding box in the unwrapped CT image for convenience.

### 2.5. CT ROI Extraction

After successfully matching the curvature images, the coordinates of the four matching points in the unwrapped CT image are calculated. These 2D coordinates and image intensity values can be converted to 3D coordinates using Equations (Equation 9) and (Equation 10), which are inverse operations of spherical unwrapping.
(9)$$\tan\left(\varphi_{i}\right)=\frac{y_{i}}{x_{i}}$$
(10)$$\sin\left(\theta_{i}\right)=\frac{z_{i}}{r_{i}}$$

To calculate $\varphi_{i}$, the corresponding width coordinate of the image is divided by the image width value, and to calculate $\theta_{i}$, the corresponding height coordinate of the image is divided by the image height value. $r_{i}$ represents the intensity of the corresponding image coordinate as shown in Equation (Equation 1) and is the distance between the origin and corresponding 3D points of CT data. These simultaneous equations are used to obtain the coordinates of the converted 3D points. Then, the 3D ROI is extracted from the 3D CT data using the four converted 3D matching coordinates. A boundary area is established based on the four converted coordinates, and the 3D point data included within this specific boundary are extracted from the 3D CT data. These extracted data represent the proper initial position for applying the ICP and are the CT ROI that will be matched to the 3D scan data. Figure 7 shows the 3D ROI extracted using the four converted 3D matching coordinates.

### 2.6. Accurate Surface Registration Using ICP Algorithm

The ICP algorithm is one of the representative ways to match different data, and in particular, it is mainly used for matching between 3D point cloud data. The ICP algorithm is based on finding the closest corresponding points between the two point clouds and minimizing the sum of the distances between them to find the correlation. Then, the data for matching are moved and rotated according to this correlation to add and match the existing data. The ICP algorithm is suitable for point cloud data registration, since it can achieve high accuracy with a simple calculation. However, when there is a significant difference between the two data sets or the matching area is small, coarse registration of the data should be performed before applying the ICP algorithm.

Coarse registration is a rough matching process that involves an approximate alignment of two data based on a proper initial location for the ICP algorithm. Using the ICP algorithm without this process can lead to convergence to a local minimum and failure of the matching process. In this study, the CT ROI, which is extracted in advance, is used instead of the entire 3D CT data to precisely match the 3D scan data using the ICP algorithm. This improves the computational speed of the ICP algorithm and makes matching more accurate because the local minimum problem is avoided at the time of matching. Figure 8 shows the results of matching two data using the ICP algorithm. The green point represents the 3D scan data, and the remainder is the CT ROI. Since the extracted CT ROI is used to provide the proper initial location for the ICP, coarse registration is not essential. However, for the stability of data matching, coarse registration is performed between two data before ICP matching. Then, as shown in Figure 8, the two 3D data are matched using the ICP algorithm.

## 3. Experimental Results and Discussion

### 3.1. Experimental Environment and Settings

The equipment utilized to obtain 3D scan data for the experiment was a 3D surface measurement sensor based on structured light, which is currently under development. This equipment comprised a projector and a camera. The principle of this 3D sensor is based on projection moire profilometry, in which a sine wave pattern is projected on the object under investigation by a projector and the object with the projected pattern is imaged by the camera. The 3D shape of the object is reconstructed by analyzing the deformation of the projected pattern in the acquired image. Detailed specifications of the surface measurement sensor are presented in Table 1. Unlike laser line scanners that use line-based scanning, this equipment scans the entire area in the field of view, allowing a wide range of 3D data to be obtained with just a few captures. Thus, it is suited to be used for image-to-patient registration in operating rooms. Furthermore, it offers advantages over other scanners in terms of accuracy and ease of use.

Figure 9 shows the experimental environment. As shown in Figure 9, a head phantom was used instead of a human head in the experiment. The 3D sensor was used to measure the surface of the natural features of the head phantom, such as eyes, nose, and ears, and the 3D CT data had the entire surface information of the head phantom. The 3D scan data consisted of approximately 20,000 points, and the 3D CT data consisted of approximately 350,000 points. The proposed algorithm was implemented using MATLAB.

### 3.2. Curvature Image Matching and CT ROI Extraction

In the proposed method, the depth image obtained using spherical unwrapping is converted into a curvature image. Then, matching points are obtained to extract the CT ROI using curvature image matching. Although the conversion process is cumbersome, it is an essential step for robust image matching using the characteristics of the curvature image. To prove this, as shown in Figure 10, image matching between depth images before conversion and image matching between curvature images were performed using NCC, respectively. The results of depth image matching are shown in Figure 10a, where matching failed due to the large difference in intensities between 3D scan data and 3D CT data. In contrast, matching performed on curvature images was successful and accurate, as shown in Figure 10b. These experimental results show the fact that curvature image matching is more robust than depth image matching.

Matching between depth images was not performed properly due to their sensitivity to changes in intensity caused by holes, rotations, and translations that occur during the unwrapping process. In contrast, curvature image matching uses calculated curvature as a relative relation of surrounding image intensity values. This means that some degree of rotation and translation, and even some deformation, can be ignored. Furthermore, by only emphasizing the targeted natural feature, robust image matching can be achieved. Figure 11 shows additional experimental results of curvature image matching. It can be seen that curvature image matching works well even if there is some deformation or rotation that can occur during the scanning process in 3D scan data.

Four matching points are obtained with curvature image matching, and these points are then converted into 3D points to extract the CT ROI based on the coordinates of the converted points. In this experiment, 30 sets of 3D scan data obtained with the 3D measurement sensor were used for CT ROI extraction. The accurate matching result of the proposed method ensures the precise extraction of the CT ROI. Part of the CT ROI extraction experimental results are shown in Figure 12.

### 3.3. Surface Registration Results Using ICP Algorithm

Image-to-patient registration is achieved by matching the 3D scan data with the extracted CT ROI. The results of an experiment performed to verify the proposed algorithm using several 3D scan data are shown in Figure 13 and Figure 14. ICP registration between the whole 3D CT data and the 3D scan data without preprocessing, such as coarse registration, failed due to the local minimum problem and scale difference, as shown in Figure 13. However, the proposed ICP registration between the CT ROI and the 3D scan data avoided the local minima during the matching process and was performed correctly, as shown in Figure 14.

Table 2 shows the ICP registration errors of the results in Figure 14. The mean ICP registration error, calculated as the average distance of all corresponding points between the two data, was about 473–840 μm, and the standard ICP registration error was about 263–804 μm. This registration error depended on the used 3D scan data.

In addition, when ICP matching was performed after manually providing appropriate initial location information for coarse registration, it took about 40 to 70 s due to the difference in scale between the two data. In contrast, the proposed method applied the ICP algorithm after extracting the CT ROI, so the ICP matching process was completed in about 6 s, and CT ROI extraction took about 11 s. Since the proposed registration was performed by converting to a similar scale, it was performed more accurately and with fewer operations.

As previously mentioned, H-K curvature can also be used for 3D surface shape classification. The 3D shape of the specific area can be distinguished using the inequality relation of the H-K curvature value. Therefore, various studies have been conducted to detect and extract natural features such as nose and eyes from 3D head data, leveraging the distinctive properties of H-K curvature [29,30]. This method can also exploit the characteristics of H-K curvature to identify natural features. Previous experiments used 3D scan data that had been taken without roll rotation when measuring the surface of the head phantom. However, by obtaining the direction vector of the natural feature region from 3D scan data using H-K classification, 3D scan data with some roll rotation can also be matched. Although some image matching algorithms that consider the template with roll rotation can be used, they take a long time to execute due to a large amount of computation [32,33,34]. In contrast, this method utilizes the already obtained H-K curvature image without the need for additional complicated operations.

Figure 15 shows the nose vector obtained by extracting the nose region, which is an elliptical convex, using the surface shape classification of H-K curvature in the curvature image of 3D scan data. The extracted nose region is represented as a binary image to simply verify the extraction. The nose vector was obtained using the maximum–minimum value and the rate of change in the intensity values of the unwrapped image corresponding to the nose region. In the unwrapped CT image, the nose vector is always vertically upward. Therefore, aligning the nose vector of the template image with this vector before NCC allows data with roll rotation to be used in the proposed algorithm. This improvement can increase the degree of freedom and convenience for the user.

Figure 16 shows the result of the ICP registration of 3D scan data that have roll rotation using the proposed vector alignment algorithm. Because the nose vector of the template image was aligned before NCC, the CT ROI was correctly extracted, and ICP registration was successfully completed. However, this has confirmed the possibility, even though it is not yet a complete algorithm, to improve the algorithm for other natural feature areas, such as the ear, and further experiments are needed in the future.

## 4. Conclusions

The study concentrated on coordinate matching between patient coordinates and image coordinates for image-to-patient registration, which is used in surgical navigation systems. Instead of using specific markers attached to the patient, the proposed method utilizes a surface measurement sensor and H-K curvature for coordinate matching to improve accuracy and convenience. The main contribution is making the image-to-patient registration process automatic. Using the proposed H-K curvature image-based registration method, the image-to-patient registration process using surface measurement data can be fully automated. Additionally, the local minimum problem of the ICP process is solved by providing a proper initial location, since the 3D ROI is extracted and used for matching, the area of data to be finally matched in ICP matching is reduced, and faster and more accurate matching results can be obtained. The proposed algorithm utilizes natural features of the patient’s face, such as eyes, nose, and ears, for matching coordinates between patient and CT data. To improve the computation speed, the 3D data are mapped to 2D depth images. Then, in order to achieve robust image matching, they are converted into H-K curvature images that emphasize the features of depth images, and image matching is performed. The three-dimensional ROI of CT data to be used for ICP matching can be obtained with the inverse operation of matched image points. Finally, the extracted CT ROI and surface measurement data are matched with the ICP algorithm. As a result of various matching experiments, in the proposed method, it is confirmed that neither of the data converge to the local minimum and that they match correctly. Further work will focus on experiments to compare the performance of the proposed method with that of traditional 3D data matching methods and to test the method on human data instead of head phantom data.

## Figures and Tables

**Figure 1 sensors-23-04903-f001:**
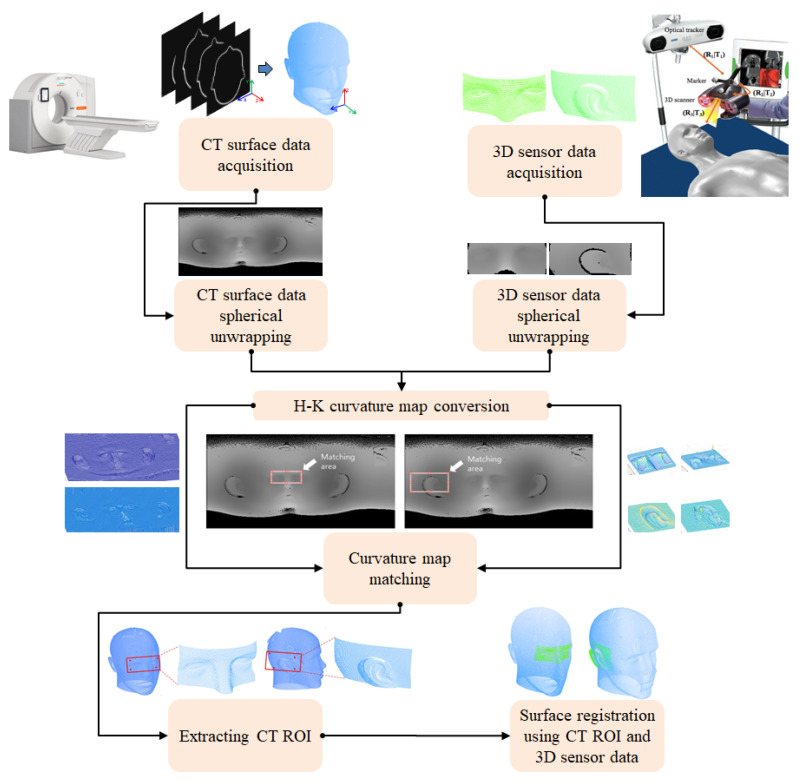
Schematic diagram of the proposed registration process.

**Figure 2 sensors-23-04903-f002:**
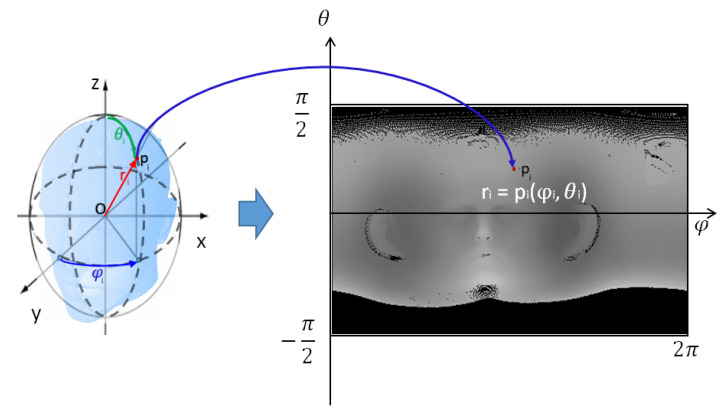
Result of mapping 3D CT data to 2D image using spherical unwrapping.

**Figure 3 sensors-23-04903-f003:**
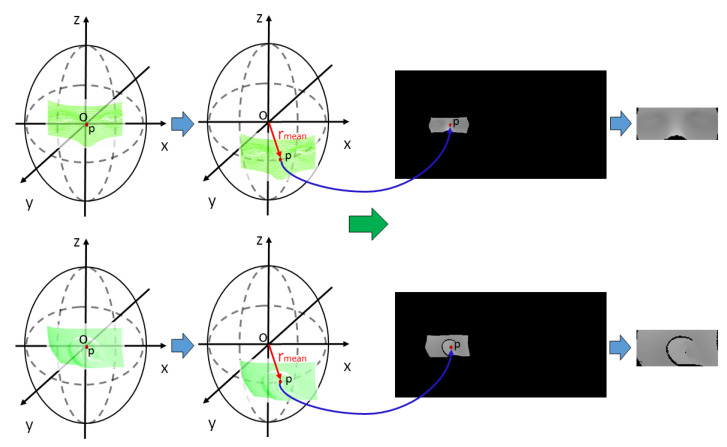
Results of mapping 3D scan data to 2D image using spherical unwrapping.

**Figure 4 sensors-23-04903-f004:**
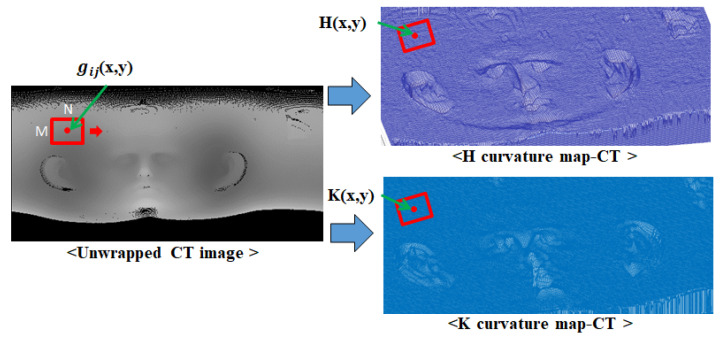
Results of curvature image conversion of the unwrapped CT image.

**Figure 5 sensors-23-04903-f005:**
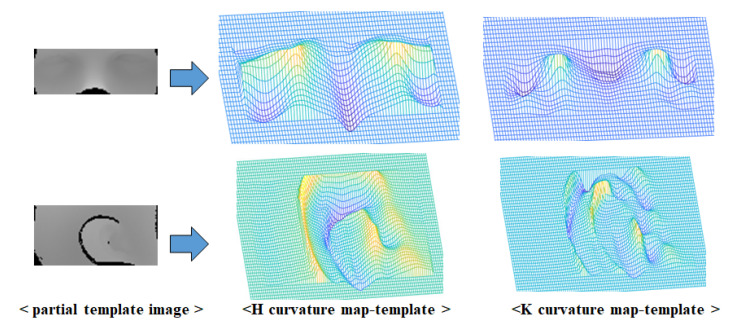
Results of curvature image conversion of the unwrapped image of 3D scan data.

**Figure 6 sensors-23-04903-f006:**
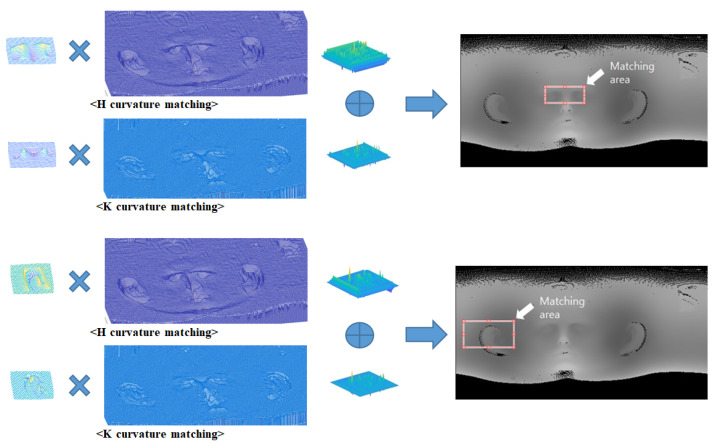
Curvature image matching using NCC.

**Figure 7 sensors-23-04903-f007:**
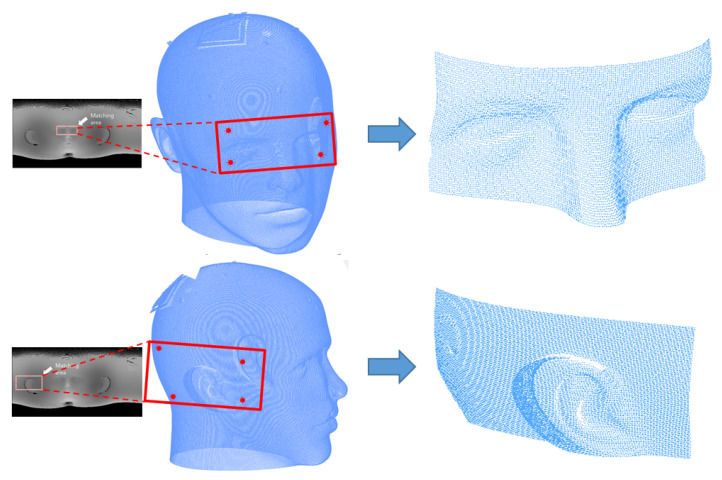
CT ROI extraction using matching points.

**Figure 8 sensors-23-04903-f008:**
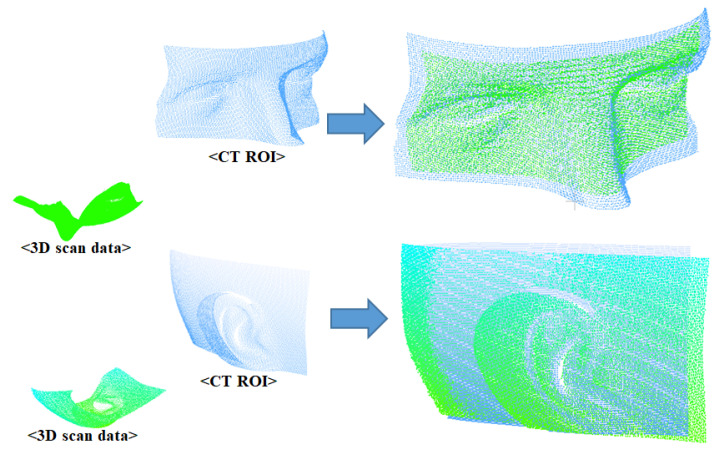
Registration results using ICP algorithm.

**Figure 9 sensors-23-04903-f009:**
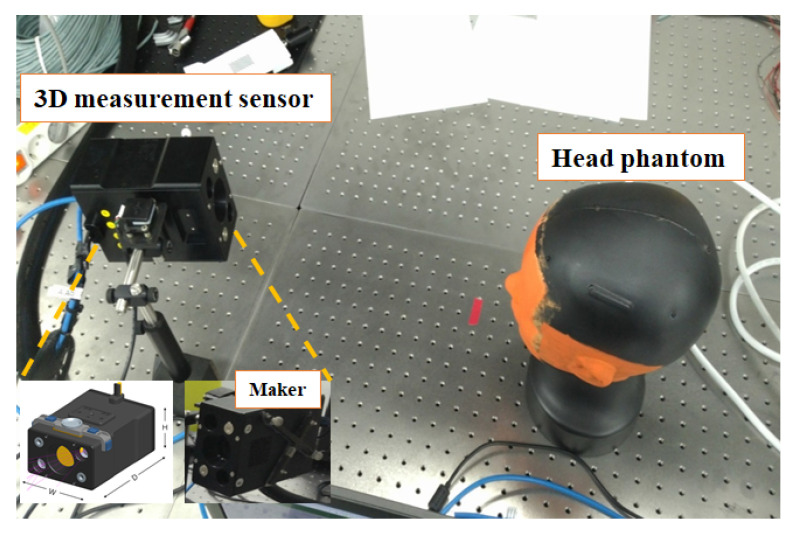
Three-dimensional measurement sensor under development and experimental environment.

**Figure 10 sensors-23-04903-f010:**
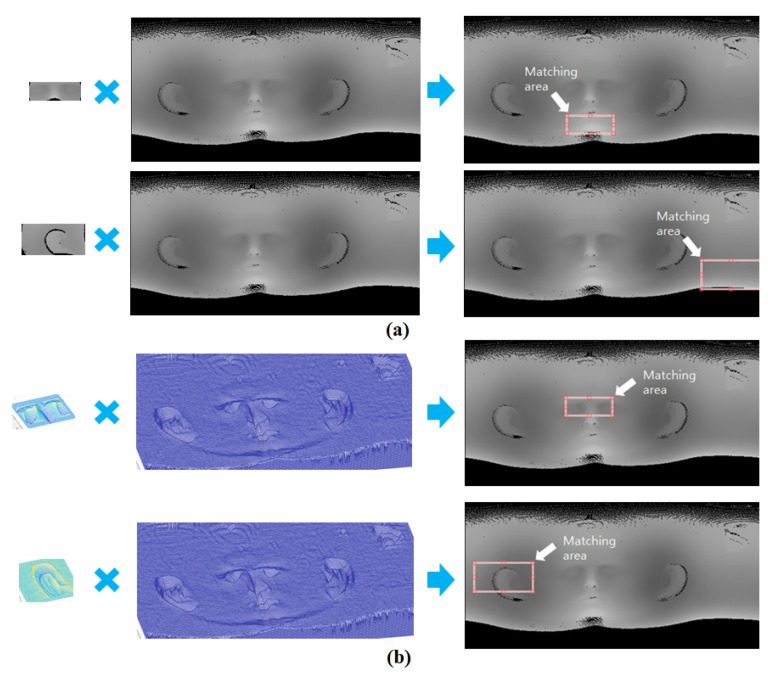
Image matching result using NCC; (**a**) depth image matching and (**b**) curvature image matching.

**Figure 11 sensors-23-04903-f011:**
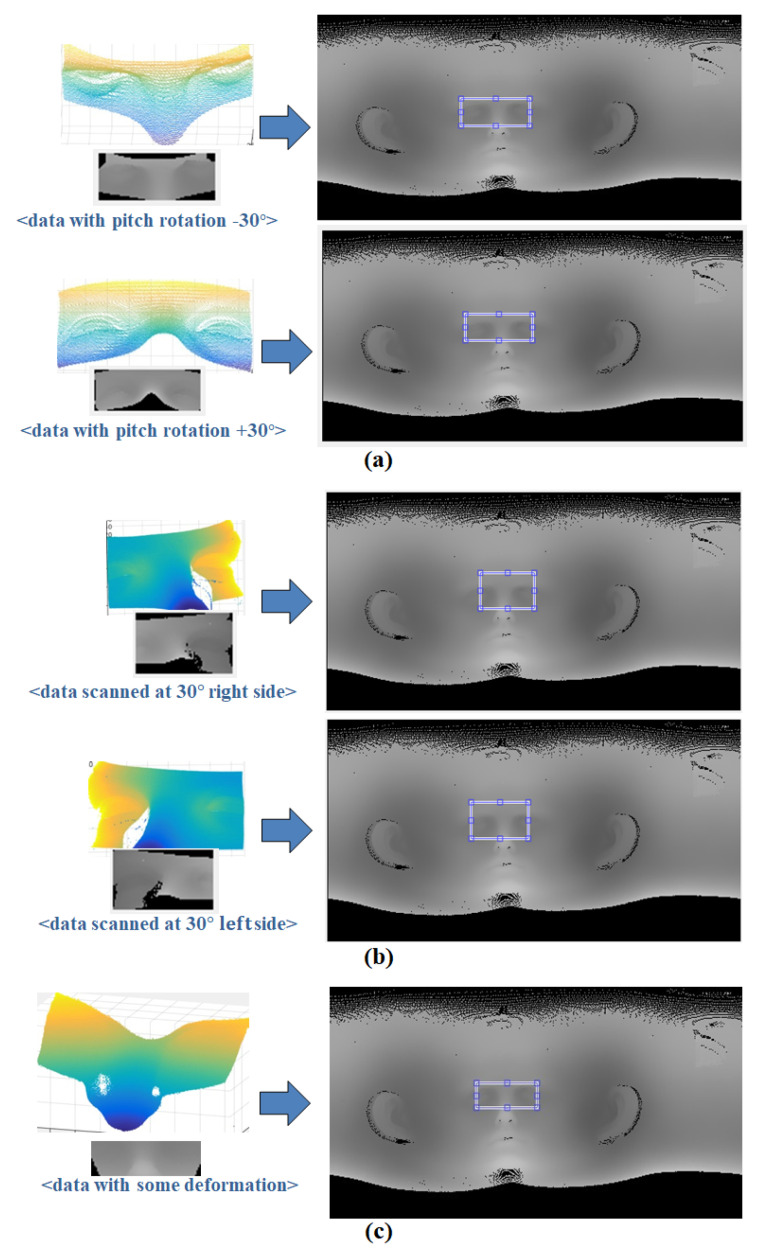
Results of curvature image matching; (**a**) using data with pitch rotation, (**b**) using data scanned on the side, and (**c**) using data with some deformation.

**Figure 12 sensors-23-04903-f012:**
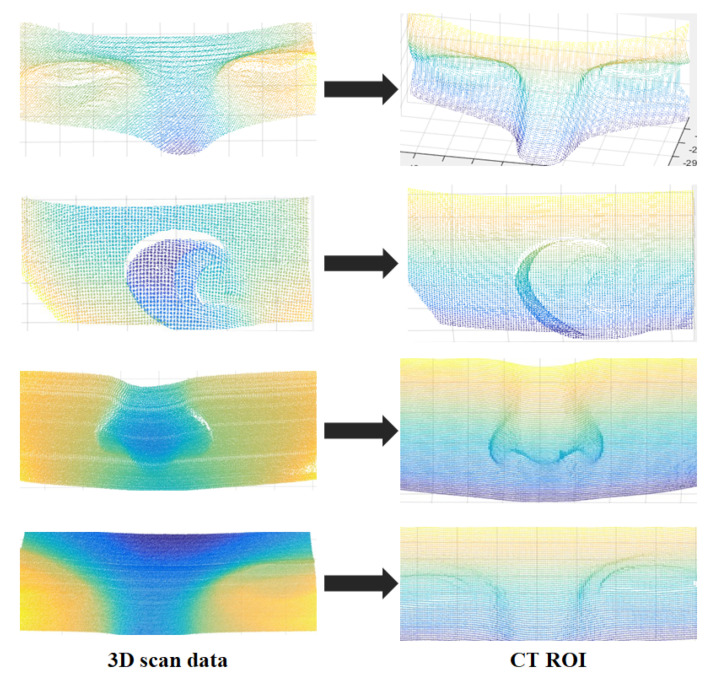
Results of CT ROI extraction.

**Figure 13 sensors-23-04903-f013:**
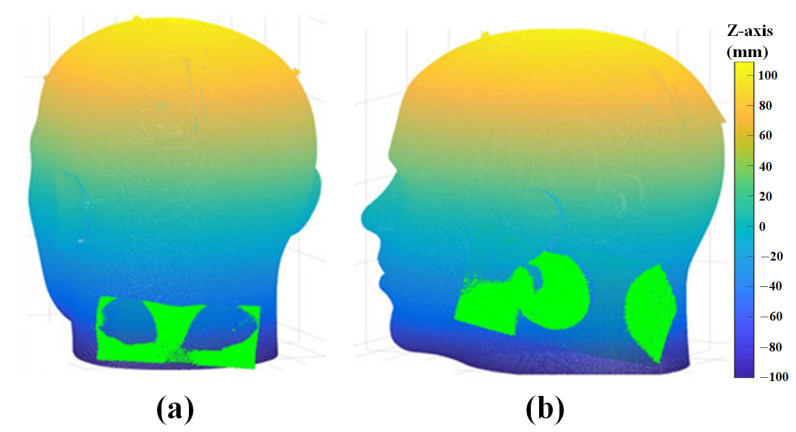
Results of ICP registration without preprocessing; (**a**) nose and eye data, and (**b**) right ear data.

**Figure 14 sensors-23-04903-f014:**
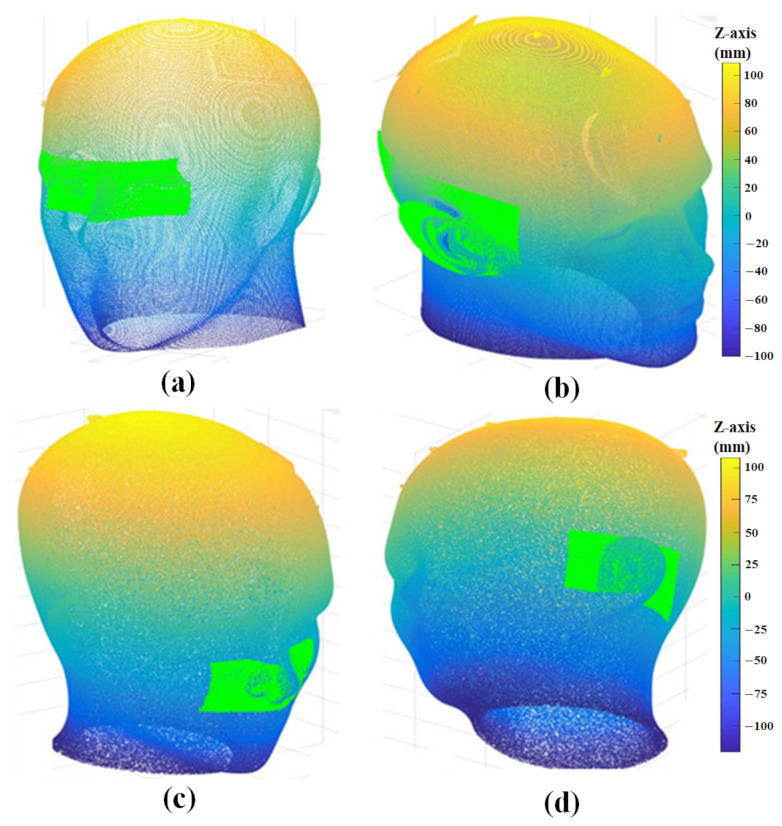
Results of proposed ICP registration; (**a**) nose and eye data, (**b**) right ear data, (**c**) data of tip of nose, and (**d**) left ear data.

**Figure 15 sensors-23-04903-f015:**
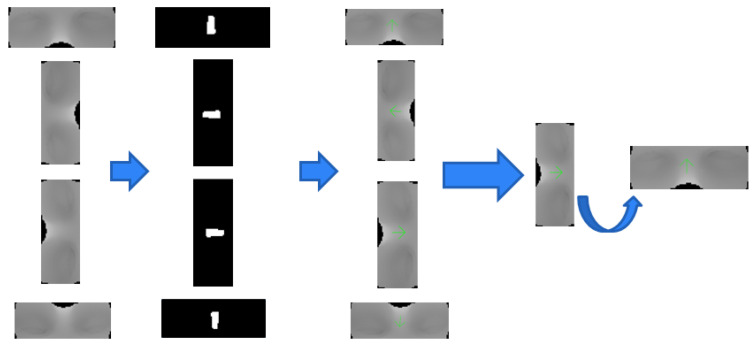
Nose region extraction and alignment using a nose vector.

**Figure 16 sensors-23-04903-f016:**
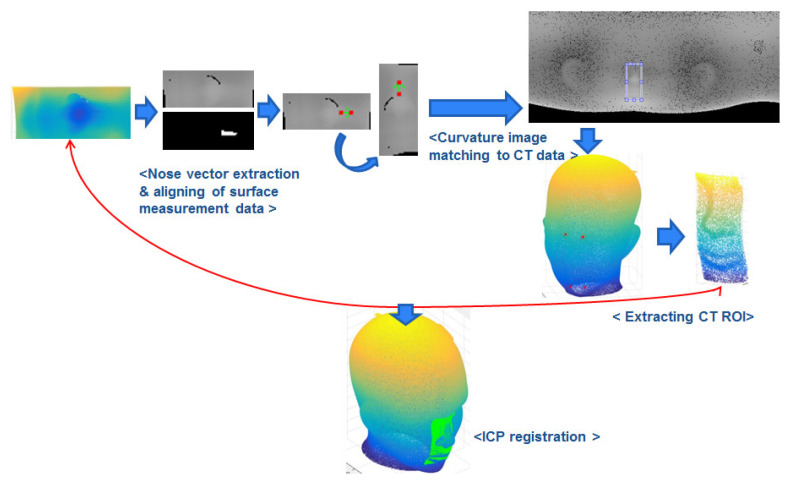
Result of proposed ICP registration using vector alignment.

**Table 1 sensors-23-04903-t001:** Specifications of the surface measurement sensor.

	Value	Unit
Resolution	2048 × 1088	pixels
Accuracy	134	μm
Measurement distance	25–35	cm
Measurement area	13 × 8	cm

**Table 2 sensors-23-04903-t002:** ICP registration errors of each result.

	Figure 14a	Figure 14b	Figure 14c	Figure 14d
Mean error	473.3 μm	834.9 μm	839.7 μm	759.1 μm
Std error	262.8 μm	803.6 μm	654.6 μm	406.8 μm

## Data Availability

Not applicable.

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
