# Peer review of "Robust H-K Curvature Map Matching for Patient-to-CT Registration in Neurosurgical Navigation Systems"

_sensors, 2023, doi:10.3390/s23104903_

Round 1

Reviewer 1 Report

The article presented describes the possible technical solutions for better alignment in CT neuronavigation procedures. The algorithm for surface matching using conversion of 3D CT data to 2D curvature images is described.

The technical solutions are described very precisely and the first experimental results are promising. The language of the article is proper and the algorithm for surface matching presents some unique solutions for the accurate alignment of the CT data to the surface area of the head.

There are no obvious flaws in the article and the proposed solutions seem sound. The incorporation of them to neuronavigational systems might be a good solution to some of the problems with accuracy in the initial image registration.

I therefore propose the article to be published in the current form.

Reviewer 2 Report

This paper introduces a markerless method for image-to-patient registration, which is the process of aligning medical images with the patient's anatomy during surgery. The method utilizes scanning data of the patient and 3D data from CT scans, and applies the iterative closest point (ICP) algorithm to register the patient's 3D surface data with the CT data. To address the limitations of the ICP algorithm, such as long convergence times and local minima issues, the authors propose a novel method that uses curvature matching to find a proper initial location for ICP. The proposed method extracts and matches the curvature information from the 3D CT data and 3D scanning data using robust curvature features that are resilient to translation, rotation, and deformation. The authors demonstrate the effectiveness of their proposed approach by accurately registering the extracted partial 3D CT data and the patient's scan data using the ICP algorithm. The results suggest that the proposed method can achieve accurate image-to-patient registration. The work is important for this field. It can be considered published on Sensors. Comments as follows:

1. Scale bar in the figures is missing. Please add a scale bar for the figures.

2. Please provide more detail about how to select the ROI, will the ROI affect the accuracy?  

3. Please format the references.

Reviewer 3 Report

very interesting job all components comply with the requirements. introduction methodology results discussion and conclusions logically related to each other. appropriate literature selected for the topic of the work
